

# Development and validation of a functional assessment tool for Chinese inpatient rehabilitation: insights from a Delphi study based on the International Classification of Functioning, Disability, and Health (ICF)

Jiahui Li[1], Guangxu Xu[2,3], Juan Jin[2], Na Li[4], Jianan Li[2] and Shouguo Liu[2]

[1] Department of Neurological Rehabilitation, The Affiliated Suzhou Hospital of Nanjing Medical University, Suzhou Municipal Hospital, Gusu School of Nanjing Medical University, Suzhou, China
[2] Center of Rehabilitation Medicine, The First Affiliated Hospital of Nanjing Medical University, Nanjing, China
[3] The Affiliated Suzhou Hospital of Nanjing Medical University, Suzhou Municipal Hospital, Gusu School of Nanjing Medical University, Suzhou, China
[4] Nanjing Yincheng Rehabilitation Hospital China, Nanjing, China

## ABSTRACT

**Objectives**. To develop and validate a functional assessment tool for inpatient rehabilitation in China using the International Classification of Functioning, Disability, and Health (ICF) Rehabilitation Set (ICF-RS) framework and the Delphi method.
**Methods**. A three-round Delphi process engaged 15 experts to refine ICF-RS items via a 5-point importance questionnaire. Validation involved 2,574 inpatients assessed with a numerical rating scale. Reliability (Cronbach's alpha) and structural validity (factor analysis) were evaluated.
**Results**. Through three rounds of Delphi meetings, 10, 2, and 1 ICF items with mean importance scores below the threshold were respectively removed, resulting in 17 ICF items achieving expert consensus for inclusion in the final assessment tool, named ICF-RS-17. Expert authority coefficient was 0.81. Cronbach's alpha exceeded 0.9. Factor analysis identified two factors explaining 68.86% (admission) and 73.25% (discharge) of variance, confirming structural validity.
**Conclusions**. The study developed a 17-item functional assessment tool, ICF-RS-17, demonstrating strong reliability and validity for inpatient rehabilitation. These findings help promote the application of the ICF in clinical settings, enhance rehabilitation clinical management, and potentially support the further development of rehabilitation insurance policies.

# INTRODUCTION

In healthcare, traditional primary medical outcome indicators, namely mortality and morbidity, are insufficient to fully capture the complexity of modern diseases (*Crimmins*

Corresponding authors
Shouguo Liu,
liushouguo2002@163.com
Jianan Li, lijianan@carm.org.cn

& Beltrán-Sánchez, 2011). Many countries, including China, are undergoing rapid demographic changes characterized by an aging population (Li et al., 2021). Concurrently, disease patterns are evolving, with a rising prevalence of chronic conditions and non-communicable diseases associated with aging and lifestyle changes (Li et al., 2021). In this context, the conventional binary outcomes of "cure" and "death" are increasingly inadequate for encapsulating the complex health trajectories and long-term functional needs of the population. Recognizing this gap, the World Health Organization (WHO) formally proposed "functioning" as a crucial third health outcome indicator alongside mortality and morbidity (Stucki & Bickenbach, 2017).

Functional assessment provides a multidimensional framework for quantifying an individual's ability to perform daily activities and participate in life roles, which is particularly vital in rehabilitation medicine, where restoring function is the primary goal (Priebe & Rintala, 1994; Levack et al., 2015). Although the International Classification of Functioning, Disability and Health (ICF) and its derivative, the ICF Rehabilitation Set (ICF-RS), offer standardized frameworks for functional assessment, their integration into routine clinical practice remains limited, especially within the specific context of inpatient rehabilitation in China (Prodinger et al., 2016a; Prodinger et al., 2016b; Liu et al., 2019; Reinhardt et al., 2016; Li et al., 2016).

In the Chinese healthcare context, where inpatient rehabilitation primarily addresses the sub-acute phase and serves patients with neurological conditions and motor impairments, certain items within the ICF-RS are not entirely applicable. For example, previous studies conducted in China have indicated that "d850 Remunerative employment" is unlikely to change during a hospital stay, and therefore, it is recommended not to include this item in a functioning score (Li et al., 2016). Furthermore, challenges persist regarding the applicability of the ICF-RS to the Chinese healthcare system, particularly concerning its role in informing rehabilitation management and insurance policies.

Therefore, there is a pressing need to develop a pragmatic, validated functional assessment tool based on the ICF framework but specifically tailored to the realities of inpatient rehabilitation settings in China. Such a tool would bridge the gap between the theoretical potential of the ICF and its practical implementation.

This study aims to develop a tailored functional assessment tool based on the ICF-RS for general inpatient rehabilitation populations in China, employing the Delphi method to identify relevant items. Additionally, it seeks to validate the reliability and validity of the tool, laying a foundation for its broader application in clinical settings.

## METHODS

### Development of a functional assessment tool using the delphi method
*Expert selection*

The Delphi method is a structured research approach employed to solicit expert opinions and establish consensus (Cai, 2022; Lee, Kim & Han, 2020). Experts were selected based on the following criteria: (1) involvement in clinical medicine, nursing, or health management; (2) holding a senior professional title or having ≥5 years of relevant experience; (3) prior

experience with the ICF. Fifteen experts from 11 hospitals across four cities in Jiangsu Province were invited to participate.

### Establishing an item pool

An item pool was established through an investigation of current clinical practices and a review of relevant literature. The research team chose to develop an item pool for a functional assessment tool by referencing the ICF-RS (*Prodinger et al., 2016a*).

### Developing the initial questionnaire

The questionnaire is structured into three distinct parts.

The first part serves as an introduction, comprising a letter to the experts, background information regarding this study, and instructions for completion.

The second part constitutes the main body of the questionnaire, which consists of an evaluation form designed to assess the items within the evaluation tool using a 5-point rating scale to gauge the importance of each item in the item pool (1 = unimportant to 5 = very important) (*Terwee et al., 2018*). The process involves three rounds of meetings in-person, with the first round allowing experts to propose additional items if they identify any suitable items missing from the initial questionnaire.

The third part includes a form for collecting expert information and a survey at the end of the questionnaire. This survey allows experts to explain their basis for judging the importance of each ICF item and their familiarity with the project. This information is essential for calculating and assessing the expert authority coefficient.

The questionnaire will be distributed on-site during the expert consensus meeting and collected immediately upon completion.

### Delphi expert consensus process

In this study, the Delphi method was utilized to identify items incorporated in the functional assessment tool based on the ICF-RS through three rounds of on-site meetings.

The criteria for selecting ICF items are as follows: (1) Remove items irrelevant to hospitalization and medical goals; (2) eliminate items that cannot be quantitatively assessed in a clinical setting; (3) retain representative items from those with similar functional dimensions; (4) exclude items for which no functional changes are expected during hospitalization.

### Evaluation indicators

The evaluation indicators include the expert positive coefficient, expert authority coefficient, mean importance scores for each ICF item, and the coordination coefficient of expert opinions (*Cai, 2022*).

The expert positive coefficient quantifies the level of participation and enthusiasm exhibited by the experts during the meeting, as indicated by the questionnaire response rate (*Lee, Kim & Han, 2020*).

The expert authority coefficient, reflecting the basis for experts' judgments concerning the content of the questionnaire and their familiarity with it (*Chen et al., 2020*). A coefficient of ≥0.70 signifies a satisfactory level of authority.

The mean importance score refers to the average rating of the importance of each item based on all returned questionnaires. In this consultation survey, a 1–5 scoring system is employed to evaluate the importance of each item in the functional assessment tool, with higher scores indicating greater importance (*He et al., 2022*). Generally, a mean importance score of ≥3.5 suggests that the item is acceptable and may be retained in the functional assessment tool.

The coordination coefficient of expert opinions is represented by the coefficient of variation (CV) for each indicator. A smaller CV indicates higher consistency in expert evaluations regarding the importance of the item, suggesting that experts' judgments about the importance of that item are more unified (*Zeng et al., 2023*). It is generally accepted that a CV ≤ 0.3 denotes an acceptable level of coordination for that indicator.

## Validation of the functional assessment tool
### Participants
Inclusion criteria: (1) Age ≥ 18 years; (2) patients receiving inpatient treatment in a rehabilitation department or rehabilitation hospital; (3) a clear disease diagnosis upon admission; (4) obtaining verbal informed consent from the patient.

Exclusion criteria: (1) Patients in critical condition or with unstable vital signs; (2) patients who are unable to undergo ICF assessment due to personal or other reasons.

This study received review and approval from the Ethics Committee of the First Affiliated Hospital of Nanjing Medical University (2020-SR-148) and has been registered with the China Clinical Trial Registration Center (No. ChiCTR2000034636).

Following the development of the functional assessment tool, we selected a total of 2,574 patients who met the aforementioned criteria from 12 rehabilitation medical institutions in Nanjing, Wuxi, and Suzhou between August 2021 and March 2023.

### Application of the assessment tool and data collection
The Numerical Rating Scale (NRS), which operates on a scale of 0 to 10, serves as the assessment method, where 0 denotes no problem and 10 signifies a complete problem. To enhance objectivity and consistency in assessments derived from the evaluation tool during practical applications and to streamline evaluators' operations, this study builds upon the Chinese grading assessment method of the ICF-RS (*Gao et al., 2018*). It further refines and develops specific evaluation criteria for each item, accompanied by a detailed evaluation manual available in the supplementary materials.

This study commissioned a relevant company to develop mobile online assessment software to facilitate data collection and unified management.

Prior to the commencement of formal data collection, evaluators from the participating research units are invited to undergo training and assessment in the ICF. Only those who pass the evaluation will be eligible to proceed to the data collection phase.

Assessing patients using the ICF at both admission and discharge in a rehabilitation inpatient setting yields a total score derived from the sum of individual item scores, which represents the overall functional status of the patients.

*Statistics and analysis*

All data analyses in this study were conducted using SPSS version 23.0.

Reliability refers to the consistency and stability of the assessment results produced by the measurement tool (*Ahmed & Ishtiaq, 2021*). This study utilized Cronbach's alpha coefficient to assess the internal consistency of the functional assessment tool, with a Cronbach's alpha value exceeding 0.75 generally regarded as indicative of good internal consistency (*He et al., 2021*).

Validity pertains to the ability of the measurement tool to accurately reflect the traits or concepts it aims to measure (*Li et al., 2022*). This study employed principal component analysis (PCA) with varimax rotation to assess the structural validity of the functional assessment tool. Prior to PCA, a Kaiser–Meyer–Olkin (KMO) test and Bartlett's test of sphericity were executed on the scores of each item. A KMO value greater than 0.9 and a *P* value from Bartlett's test of less than 0.001 suggest that PCA can be conducted. The number of components (factors) to retain was determined based on eigenvalues >1.0. To enhance the interpretability of the component structure and achieve a simple structure, the extracted components were subjected to orthogonal rotation using the varimax method. Items were considered to load significantly on a component if their rotated factor loading was ≥|0.40|. Items exhibiting cross-loadings (loading ≥|0.40| on more than one component) were assigned to the component that aligned most closely with their conceptual meaning and clinical relevance within the ICF framework and inpatient rehabilitation context, as determined by the research team.

# RESULTS

## Developing a functional assessment tool using the Delphi method
### Basic information about the experts

In this study, three iterative consensus rounds were conducted with 15 experts, all of whom participated in every session. Participants possessed a minimum of five years of clinical experience or senior professional titles, ensuring field proficiency. The experts were recruited from 11 hospitals across four cities in Jiangsu Province. Further details are provided in Table 1.

### The first round of the Delphi process

Figure 1 summarizes the Delphi process for developing the functional assessment scale. In the first round of the survey, the expert positive coefficient was 100%. A total of 10 ICF items were found to have a mean importance score <3.0 (Table S1). These items were subsequently removed and excluded from further rounds (b640 Sexual functions, d240 Handling stress and other psychological demands, d415 Maintaining a body position, d420 Transferring oneself, d455 Moving around, d470 Using transportation, d660 Assisting others, d770 Intimate relationships, d850 Remunerative employment, d920 Recreation and leisure). A total of 5 ICF items had a mean importance score ≥3.0 and <3.5, and these items were selected for detailed discussion in the next round (b710 Mobility of joint functions, d465 Moving around using equipment, d520 Caring for body parts, d570 Looking after one's health, d640 Doing housework). Among the items with a mean score ≥3.5, one item

**Table 1 Basic information about the experts.**

|  | Basic information | n | Percentage (%) |
|---|---|---|---|
| Years of service | 10–19 years | 3 | 20.00 |
|  | 20–29 years | 5 | 33.33 |
|  | 30–39 years | 4 | 26.67 |
|  | 40–49 years | 2 | 13.33 |
|  | 50–59 years | 1 | 6.67 |
| Education | Undergraduate | 5 | 33.33 |
|  | Master's degree | 4 | 26.67 |
|  | Doctoral candidate | 6 | 40.00 |
| Title | Senior | 11 | 73.33 |
|  | Associate senior | 4 | 26.67 |
| Occupation | Clinical physician | 9 | 60.00 |
|  | Nursing staff | 4 | 26.67 |
|  | Rehabilitation therapist | 1 | 6.67 |
|  | Healthcare administrative personnel | 1 | 6.67 |

exhibited a CV > 0.3, indicating a lack of consensus among the experts. This item was also marked for detailed discussion in the next round. No new items suitable for the assessment tool, not included in the initial questionnaire, were identified during this round.

### The second round of the Delphi process

The second round of the questionnaire included items from the first round with a mean importance score of ≥3.0. Items scoring <3.0 in the first round were excluded from subsequent surveys. Items scoring ≥3.0 and <3.5 or with a CV >0.3 in the first round were the focus of discussion in this round.

In the second round, the expert positive coefficient remained 100%. Two ICF items had a mean importance score <3.0 and were directly removed (Table S2, d570 Looking after one's health, d640 Doing housework). Three items scored ≥3.0 and <3.5 and remained under discussion for the next round due to their CV >0.3, which indicated insufficient consensus (b710 Mobility of joint functions, d465 Moving around using equipment, d520 Caring for body parts). All other items had a mean score of ≥3.5 and a CV ≤0.3.

### The third round of the Delphi process

The third round included items from the second round with a mean importance score of ≥3.0. Items scoring <3.0 in the second round were excluded. Items with a score ≥ 3.0 and <3.5 or a CV >0.3 were discussed in detail.

In the third round, the expert positive coefficient was 100%. This round aimed to finalize the items for inclusion in the functional assessment tool. One item had a mean importance score <3.5 and did not meet the inclusion criteria, leading to its exclusion (Table S3, d520 Caring for body parts). Two items achieved mean importance scores of ≥3.5 but had CV >0.3, reflecting insufficient consensus (b710 Mobility of joint functions, d465 Moving around using equipment). Nevertheless, considering the need to cover as many functional dimensions as possible, these two items were retained. All other items

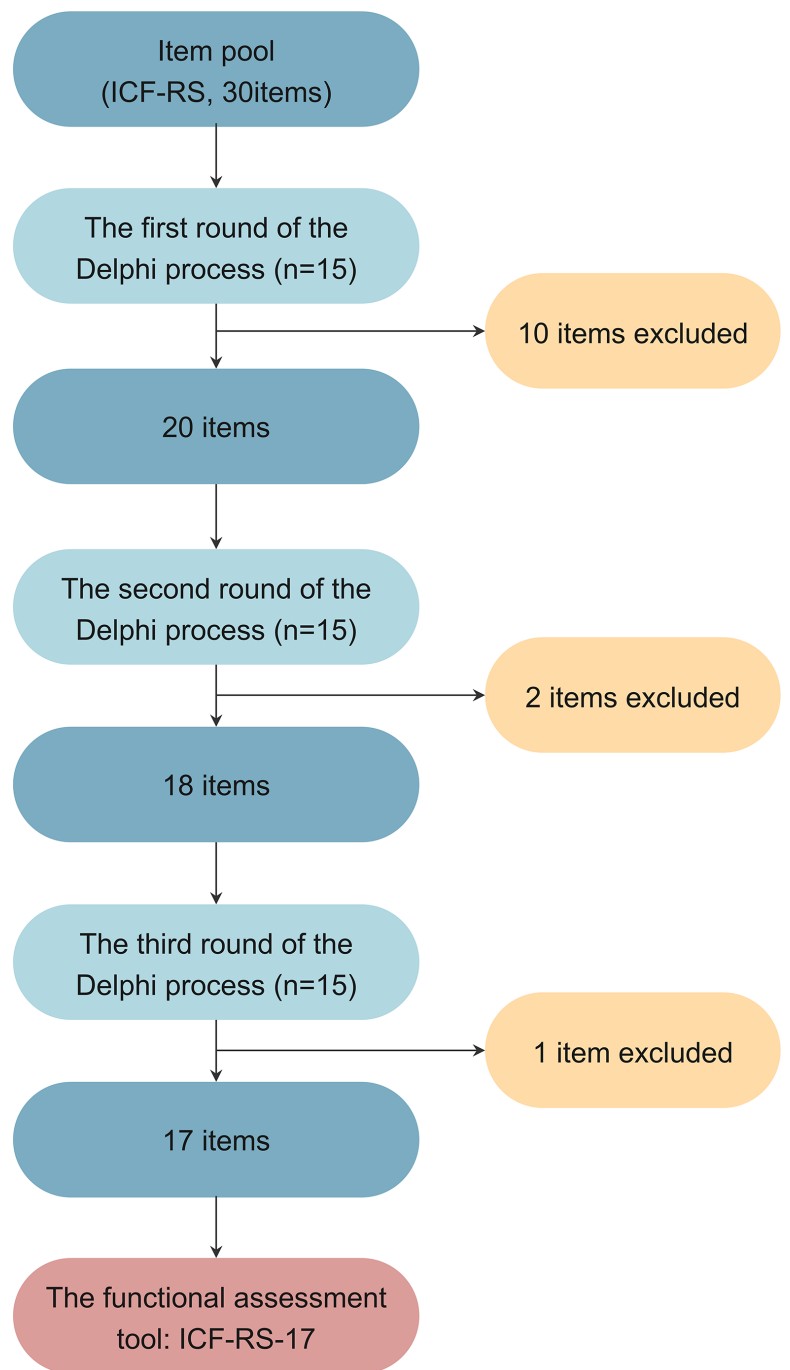

**Figure 1** The Delphi process of tool development.

scored ≥3.5 with a CV ≤0.3. After three rounds, 17 ICF items met the inclusion criteria and achieved consensus among the experts. These items are presented in Table 2. Finally, the functional assessment tool is named ICF-RS-17.

**Table 2    Results of the final included items in the third round of the survey.**

| ICF item | Numbers of individuals for each importance score | | | | | Mean | CV |
|---|---|---|---|---|---|---|---|
| | 5 | 4 | 3 | 2 | 1 | | |
| b130 Energy and drive functions | 12 | 2 | 0 | 0 | 0 | 4.857 | 0.075 |
| b134 Sleep functions | 12 | 1 | 1 | 0 | 0 | 4.786 | 0.121 |
| b152 Emotional functions | 8 | 6 | 0 | 0 | 0 | 4.571 | 0.112 |
| b280 Sensation of pain | 12 | 2 | 0 | 0 | 0 | 4.857 | 0.075 |
| b455 Exercise tolerance functions | 9 | 4 | 1 | 0 | 0 | 4.571 | 0.141 |
| b620 Urination functions | 13 | 1 | 0 | 0 | 0 | 4.929 | 0.054 |
| b710 Mobility of joint functions | 4 | 4 | 3 | 1 | 2 | 3.500 | 0.400 |
| b730 Muscle power functions | 6 | 7 | 1 | 0 | 0 | 4.357 | 0.145 |
| d230 Carrying out daily routine | 8 | 3 | 3 | 0 | 0 | 4.357 | 0.193 |
| d410 Changing basic body position | 9 | 4 | 1 | 0 | 0 | 4.571 | 0.141 |
| d450 Walking | 13 | 1 | 0 | 0 | 0 | 4.929 | 0.054 |
| d465 Moving around using equipment | 5 | 5 | 0 | 2 | 2 | 3.643 | 0.412 |
| d510 Washing oneself | 11 | 3 | 0 | 0 | 0 | 4.786 | 0.089 |
| d530 Toileting | 9 | 5 | 0 | 0 | 0 | 4.643 | 0.107 |
| d540 Dressing | 11 | 2 | 0 | 0 | 1 | 4.571 | 0.238 |
| d550 Eating | 13 | 1 | 0 | 0 | 0 | 4.929 | 0.054 |
| d710 Basic interpersonal interactions | 5 | 8 | 1 | 0 | 0 | 4.286 | 0.143 |

### *Expert authority coefficient*

The experts' average familiarity with the item content during this consensus meeting was 0.74, and their average judgment basis was 0.88. The final calculated authority coefficient was 0.81, indicating a strong level of authority in the results.

## Validation of reliability and validity

### *Reliability analysis*

Using the ICF-RS-17, data were collected from rehabilitation inpatients.

Prior to calculating Cronbach's α coefficient, the item-total correlations and inter-item correlations were examined to ensure they were positive and of moderate magnitude, satisfying a key assumption for internal consistency reliability.

Statistical analysis showed a Cronbach's α coefficient of 0.950 for admission assessments and 0.961 for discharge assessments, both exceeding 0.9, indicating high reliability. Split-half reliability coefficients were 0.899 (admission) and 0.922 (discharge), demonstrating strong internal consistency. These findings confirm that the measurement results of this tool are reliable and stable. The classification of ICF items is presented in Table 3.

### *Validity analysis*

KMO testing and Bartlett's test of sphericity were conducted. For the admission assessment, the KMO value was 0.955, with a Bartlett's test statistic of 40,745.826 ($P < 0.001$). For the discharge assessment, the KMO value was 0.958, with a Bartlett's test statistic of 47,309.846 ($P < 0.001$). These results supported exploratory factor analysis for structural validity assessment.

**Table 3** Split-half reliability of the ICF-RS-17 (*n* = 2574).

| | Included items | Cronbach's α coefficient (admission) | Split-half reliability (discharge) | Cronbach's α coefficient (admission) | Split-half reliability (discharge) |
|---|---|---|---|---|---|
| Part 1 | d450, b620, b730, d410, d530, d540, d710, b134, b280 | 0.902 | | 0.924 | |
| Part 2 | b455, b710, d230, d465, d510, d550, b130, b152 | 0.897 | 0.899 | 0.920 | 0.922 |

In addition to conducting the KMO test for sampling adequacy and Bartlett's test of sphericity, we evaluated several assumptions relevant to exploratory factor analysis. Our sample size (*N* = 2,574) significantly surpasses the commonly recommended thresholds. Each item was measured using a 0–10 NRS and treated as a continuous variable. Visual inspections of scatterplot matrices suggested generally linear relationships among the items. Moreover, the absence of extreme multicollinearity was confirmed, as indicated by the results of our multicollinearity assessment.

Principal component analysis with maximum variance revealed two factors with eigenvalues >1 for both assessments. Cumulative variance explained was 68.861% (admission) and 73.245% (discharge). Factor loadings of items ≥0.4 were analyzed. The rotated factor loadings (using varimax rotation) for each item are presented in Table 4.

Based on the pattern of high loadings (≥|0.40|), the components were interpreted and labelled as follows:

Component 1: Self-Care and Activities - This component comprised items primarily related to basic mobility, self-care tasks, and daily activities.

Component 2: Emotion and Mental Health - This component encompassed items primarily associated with emotional states, cognitive functions, and interpersonal interactions.

Two items, b620 Urination functions and b130 Energy and drive functions, exhibited significant loadings (≥|0.40|) on both components (Table 4). After careful consideration of their conceptual content within the ICF framework and their relevance to inpatient rehabilitation goals:

b620 Urination functions was assigned to Component 1 (Self-Care and Activities) as it pertains more directly to a fundamental bodily function impacting daily activities and self-care independence.

b130 Energy and drive functions was assigned to Component 2 (Emotion and Mental Health) as it reflects motivational states and energy levels, which are conceptually closer to emotional and mental well-being, and can significantly influence engagement in interpersonal interactions.

## DISCUSSION

This study developed and validated the ICF-RS-17, providing a foundational reference for clinical application.

**Table 4 Factors, items and factor loadings of the ICF-RS-17 ($n = 2574$).**

| Items | Factor loadings (admission) | | Intercommunity (admission) | Factor loadings (discharge) | | Intercommunity (discharge) |
|---|---|---|---|---|---|---|
| | Factor 1 | Factor 2 | | Factor 1 | Factor 2 | |
| d530 | 0.919[*] | 0.130 | 0.862 | 0.918[*] | 0.164 | 0.869 |
| d540 | 0.905[*] | 0.159 | 0.845 | 0.912[*] | 0.181 | 0.865 |
| d410 | 0.899[*] | 0.213 | 0.853 | 0.916[*] | 0.217 | 0.885 |
| d510 | 0.888[*] | 0.114 | 0.802 | 0.891[*] | 0.152 | 0.817 |
| d465 | 0.886[*] | 0.137 | 0.804 | 0.887[*] | 0.124 | 0.802 |
| d230 | 0.884[*] | 0.168 | 0.810 | 0.890[*] | 0.124 | 0.862 |
| d450 | 0.870[*] | 0.074 | 0.763 | 0.912[*] | 0.176 | 0.808 |
| b455 | 0.778[*] | 0.281 | 0.684 | 0.794[*] | 0.285 | 0.712 |
| d550 | 0.733[*] | 0.308 | 0.632 | 0.773[*] | 0.274 | 0.673 |
| b730 | 0.726[*] | 0.247 | 0.588 | 0.807[*] | 0.230 | 0.704 |
| d710 | 0.636[*] | 0.387 | 0.554 | 0.710[*] | 0.318 | 0.605 |
| b620 | 0.590[*] | 0.416[*] | 0.521 | 0.641[*] | 0.351 | 0.534 |
| b130 | 0.577[*] | 0.509[*] | 0.592 | 0.632[*] | 0.491[*] | 0.641 |
| b710 | 0.409[*] | 0.359 | 0.296 | 0.566[*] | 0.285 | 0.401 |
| b134 | 0.181 | 0.845[*] | 0.747 | 0.191 | 0.882[*] | 0.814 |
| b152 | 0.202 | 0.839[*] | 0.745 | 0.255 | 0.841[*] | 0.772 |
| b280 | 0.002 | 0.779[*] | 0.606 | 0.102 | 0.823[*] | 0.687 |

**Notes.**
[*]Factor loading > 0.40.

The Delphi method is a structured research approach used to gather expert opinions and achieve consensus (*Lee, Kim & Han, 2020*). This methodology, widely utilized in creating ICF-based assessment tools and standards, ensures the tool is tailored to specific objectives, enhancing its applicability and relevance to the target population (*De Wind et al., 2022*; *Zhang et al., 2024*; *Hernandez-Lazaro et al., 2023*).

The finalized tool comprises 17 items, with eight from the "b Body functions" domain of the ICF framework. Initially, the domain included nine items, but "b640 Sexual functions" was excluded after the first Delphi round due to its limited relevance and feasibility in inpatient settings. The remaining items, such as "b710 Mobility of joint functions" and "b730 Muscle power functions," reflect common rehabilitation goals (*Levack et al., 2015*). Experts agreed that "b640 Sexual functions" was challenging to assess and not directly tied to hospitalization goals, leading to its removal.

The remaining nine items belong to the "d Activities and participation" domain. Within the "d2 General tasks and demands" category, "d230 Carrying out daily routine" was retained, while "d240 Handling stress and other psychological demands" was excluded due to its limited relevance to inpatient care. In the "d4 Mobility" category, three items, "d410 Changing basic body position", "d450 Walking", and "d465 Moving around using equipment", were retained, while others were removed to minimize dimensional overlap and enhance practicality. Similarly, in the "d5 Self-care" category, items such as "d510 Washing oneself" and "d550 Eating" were retained, while broader descriptions and less

quantifiable items, or those that are poorly relevant to inpatient medical goals,while broader and less quantifiable items like "d520 Caring for body parts" were excluded. Items from categories such as "d6 Domestic life", "d8 Major life areas" and "d9 Community, social, and civic life" were excluded, as these activities are typically beyond the scope of inpatient care. However, "d710 Basic interpersonal interactions" was retained as a representative item to evaluate patients' interpersonal functioning.

The initial item pool excluded the "e Environmental factors" and "s Body structures" domains of the ICF framework. Although some studies suggest that environmental factors are crucial in rehabilitation settings, experts at the conference argued that modifying these factors in a targeted manner during hospitalization is often challenging within the context of inpatient rehabilitation in China, rendering them ineffective as indicators for medical assessment (*Saverino et al., 2015*; *O'Halloran, Worrall & Hickson, 2011*; *Della Vecchia et al., 2023*). Moreover, body structures are rarely included in relevant studies (*Nuño et al., 2018*; *Wildeboer, Stallinga & Roodbol, 2022*), and achieving quantifiable changes within a brief hospitalization period is similarly difficult. Additionally, body structures often manifest as body functions, making body functions significantly more relevant for clinical outcome evaluations focused on functional assessment. These exclusions align with clinical priorities, emphasizing body functions and activities directly related to functional outcomes.

The 0–10 NRS was adopted for scoring the ICF-RS-17. Although the 5-point Likert scale is internationally recognized, its ordinal nature and uneven value intervals can complicate practical application and statistical analysis (*Chang et al., 2012*; *Rauch, Cieza & Stucki, 2008*). In contrast, the NRS, widely used in clinical practice, allows for straightforward quantification and interpretation (*Liu et al., 2019*; *Li et al., 2022*; *Gimigliano et al., 2019*). The scoring system for the ICF-RS-17 ranges from 0 to 170, with lower scores indicating better functional status.

Reliability and validity assessments confirmed the tool's robustness (*Liu et al., 2019*). Reliability, defined as the consistency and stability of measurements, was high in both admission and discharge evaluations (*Ahmed & Ishtiaq, 2021*). Validity analyses further demonstrated the tool's alignment with its conceptual framework, classifying the 17 items into two categories: "Self-Care and Activities" and "Emotion and Mental Health" (*Pieber et al., 2015*). These classifications underscore the tool's construct validity and its ability to capture key aspects of inpatient functional assessment (*Li et al., 2022*; *Mantzalas et al., 2024*).

Unlike traditional evaluation systems that prioritize mortality and organic outcomes, the ICF-RS-17 emphasizes holistic patient function and quality of life. A previous study showed that the use of ICF enhances communication and improves rehabilitation outcomes (*Stucki et al., 2017*). The ICF framework fosters interdisciplinary collaboration, ensuring coordinated care among healthcare providers from different disciplines. Thus, its application extends beyond rehabilitation departments, offering utility across various inpatient settings.

Despite its strengths, the study has limitations. Most of the consulted experts were from rehabilitation specialties, potentially limiting the tool's adaptability to other clinical contexts. Given the tool's primary design objective of achieving broad functional coverage

with minimal items, this approach inherently entails certain limitations. Consequently, some domains also critical to comprehensive rehabilitation assessment, such as the functional rehabilitation, the Activities and Participation domain (ICF), and the overall quality of life metrics, were excluded from the final item set. On the other hand, the relatively high Cronbach's alpha coefficient observed in this study suggests that there may be redundancy among the items included in the assessment tool. Future research could refine the tool to broaden its clinical applicability.

## CONCLUSION

In conclusion, this study employed the Delphi method to develop the ICF-RS-17, a functional assessment tool comprising 17 ICF items. Statistical analyses confirmed its reliability and validity, offering a valuable resource for clinical practice. The ICF-RS-17 holds potential to advance rehabilitation management and inform policy development in rehabilitation insurance.

## ACKNOWLEDGEMENTS

The authors would like to express their gratitude to 12 rehabilitation medical institutions for their support in the implementation of the project.

### Funding
This study is supported by the Wuxi Healthcare Security Bureau and Nanjing Healthcare Security Bureau. The funders had no role in study design, data collection and analysis, decision to publish, or preparation of the manuscript.

### Grant Disclosures
The following grant information was disclosed by the authors:
Wuxi Healthcare Security Bureau and Nanjing Healthcare Security Bureau.

### Competing Interests
The authors declare there are no competing interests.

### Author Contributions
- Jiahui Li performed the experiments, analyzed the data, prepared figures and/or tables, authored or reviewed drafts of the article, and approved the final draft.
- Guangxu Xu conceived and designed the experiments, analyzed the data, authored or reviewed drafts of the article, and approved the final draft.
- Juan Jin performed the experiments, authored or reviewed drafts of the article, and approved the final draft.
- Na Li performed the experiments, authored or reviewed drafts of the article, and approved the final draft.

- Jianan Li conceived and designed the experiments, authored or reviewed drafts of the article, and approved the final draft.
- Shouguo Liu conceived and designed the experiments, analyzed the data, authored or reviewed drafts of the article, and approved the final draft.

## Human Ethics

The following information was supplied relating to ethical approvals (i.e., approving body and any reference numbers):

The Ethics Committee of the First Affiliated Hospital of Nanjing Medical University

## Data Availability

The raw data is available in the Supplemental File.

## Supplemental Information

Supplemental information for this article can be found online at http://dx.doi.org/10.7717/peerj.20280#supplemental-information.

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
