# Peer review of "Development and validation of a functional assessment tool for Chinese inpatient rehabilitation: insights from a Delphi study based on the International Classification of Functioning, Disability, and Health (ICF)"

_PeerJ, doi:10.7717/peerj.20280_

## Round 0.1 · original submission · Major Revisions

· Academic Editor

Major Revisions

·

Basic reporting

The reviewed paperwork is clear and written in professional English. The authors' idea is commendable and important to pursue.
The structure adheres to journal standards.
The abstract summarizes the entire work. The document is concise and includes the rationale for the work, methodology, results, and conclusions.
The introduction is clear and provides enough background.
The references from the medical literature are pertinent and current.
Figures and tables are relevant, well-labeled, and described.

Experimental design

This original research aligns with the journal's scope.
The research question is clear and relevant. The authors describe a Delphi exercise to develop and validate a functional assessment tool for inpatient rehabilitation in China using the ICF Rehabilitation Set (ICF-RS) framework. They correctly identify a knowledge gap regarding the application's use for rehabilitation inpatients.
The work is rigorous and adheres to scientific and ethical standards. The text is correct and suitable.
The methods are detailed, limitations are addressed, and results are clear. The statistical analysis confirms the reliability and validity of the final item list.

Validity of the findings

The conclusions are clearly articulated and connected to the original research question. The 17-item functional assessment tool, ICF-RS-17, is a valid instrument for applying the ICF in clinical settings.

Additional comments

It is recommended that this paper be published as it is.

·

Basic reporting

The manuscript is clearly written and professionally structured, with an introduction and background that provide sufficient context. The references cited are relevant and up to date, supporting the rationale and framework of the study. Figures and tables are appropriately labeled and well described, enhancing the clarity of the presented results. The inclusion of raw data and a transparent account of the tool development process further strengthens the manuscript. Minor language polishing is recommended to improve consistency and flow, particularly with regard to verb tense usage and phrasing.

Experimental design

The study addresses a well-defined and clinically relevant research question, effectively filling an important knowledge gap in rehabilitation assessment within the Chinese context. The use of the Delphi method, with clearly defined inclusion criteria for expert participants, is both appropriate and well implemented. Methodological rigor is evident, particularly in the pre-specification and justification of item retention and removal criteria. Ethical approval has been obtained, and informed consent procedures are adequately described.

Validity of the findings

The reliability analysis demonstrates strong internal consistency, with Cronbach’s alpha values exceeding 0.9 and robust split-half reliability results. The validity of the tool is supported by KMO and Bartlett’s test outcomes, confirming the data’s suitability for factor analysis. The resulting two-factor solution is well justified and aligns with the conceptual structure of the tool. Items showing cross-loadings (e.g., b130 and b620) are thoughtfully categorized based on clinical relevance. Overall, the conclusions are well supported by the data and remain appropriately confined to the scope of the study.

Additional comments

The study has several notable strengths. It benefits from a large sample size (n = 2,574) drawn from multiple institutions, enhancing the generalizability of the findings. The Delphi process was executed with high participation and employed clear decision-making criteria, contributing to the rigor of the tool’s development. The resulting ICF-RS-17 is well aligned with the functional needs of patients in inpatient rehabilitation settings in China. Furthermore, the tool holds significant clinical and policy relevance, supporting the broader implementation of functional assessment practices.
Nevertheless, I have suggestions for improvements:
Introduction
1. You should clearly state that this study focuses on inpatient rehabilitation during the sub-acute phase. Currently, this is not sufficiently clear. While you refer to "functional evaluations," it is important to acknowledge that during the sub-acute phase, rehabilitation goals are typically more impairment-based (aligned with the Body Functions and Structures component of the ICF), rather than focused on Activities and Participation, which are more relevant in the chronic phase. This rationale underpins the exclusion of items related to long-term or chronic rehabilitation.
2. You also need to specify the target patient population more precisely. Are you referring to general rehabilitation inpatients, or is this study specific to neurorehabilitation? Based on the selected items and context, it appears that the focus is on patients with motor impairments. However, this should be stated explicitly, including whether patients with communication, cognitive, swallowing, or mental health impairments were included or excluded.
Methods
1. Expert Selection
Line 95: Please specify the clinical disciplines of the experts. Did the panel include rehabilitation clinicians such as physiotherapists, occupational therapists, speech and language therapists, or physicians? A table summarizing the demographic and professional backgrounds of the expert panel would greatly enhance transparency.
Line 96: Clarify the term “associate senior title.” Does this refer to a specific professional or academic rank? Please define it for an international audience.
Line 97: Define what is meant by “prior experience with the ICF.” Did this include clinical use, research, training, or certification?
2. Establishing the Item Pool
Line 101: Please explain in detail how you investigated current clinical practices. For example, did you conduct interviews, surveys, site visits, or chart reviews?
Line 102: Please provide more information on the literature review process. Which databases did you search, what were your inclusion/exclusion criteria, and how many articles were reviewed?
Line 102: Clarify the criteria used to determine which ICF items were included in the initial item pool. On what basis did you decide they were relevant?
3. Developing the Initial Questionnaire
Line 112: Please describe how the three rounds of the Delphi process were conducted. Were the meetings held in person or online? Did all experts participate in each round?
4. Participation Rates
Please clarify the percentage of expert panel participation across the three Delphi rounds. Were all 15 experts present for all rounds, or did participation vary? If so, provide exact numbers or percentages.
Discussion
It is important to acknowledge, particularly in the Limitations section, that many items related to functional rehabilitation, the Activities and Participation domain (ICF), and overall quality of life were excluded from the final tool. This is a significant limitation, as it may restrict the tool’s comprehensiveness in capturing broader aspects of functioning relevant to inpatient rehabilitation. This should be explicitly discussed in the limitations.

Reviewer 3 ·

Basic reporting

Not clear.
Lack of literature with no justification for the conduct of this research
the last flow diagram need to be modified
I need to see the list of items excluded and their reason for exclusion from the experts.
I also need to see the permission from ICF-RS to modify their questionnaire (30 items).

Experimental design

The research question is not clearly stated
replication of the study is questionable
permission from authority is not clearly stated
I suspect a salami publication, as an RCT reference is provided.

Validity of the findings

The prinicpal factor analysis was not explained well
assumption test and how missing values were handled was not mentioned

Additional comments

All the detailed comments are attached as annotated manuscript file

Annotated reviews are not available for download in order to protect the identity of reviewers who chose to remain anonymous.

---

## Round 0.2 · accepted · Accept

· Academic Editor

Accept

Thank you for revising your manuscript to address the reviewers' concerns. Reviewer 2 now recommends acceptance and I am satisfied that the other reviewers' comments have been addressed. The manuscript is now ready for publication.